# Spatio-temporal Evolution of Wet-Dry Event Features and their Transition across Upper Jhelum Basin (UJB)-South Asia

Rubina Ansari[1], Giovanna Grossi[1]

[1] Department of Civil, Environmental, Architectural Engineering and Mathematics, University of Brescia, Italy.

Correspondence to: Rubina Ansari (r.ansari@unibs.it)

## Abstract

The increasing rate of occurrence of extreme events (droughts and floods) and their rapid transition magnifies the associated socio-economic impacts with respect to those caused by the individual event. Understanding of spatio-temporal evolution of wet-dry events collectively, their characteristics and transition (wet to dry and dry to wet) is therefore significant to identify and locate most vulnerable hotspots, providing the basis for the adaptation and mitigation measures. The Upper Jhelum Basin (UJB)-South Asia was selected as a case study, where the relevance of wet-dry events and their transition have not been assessed yet, despite of clear evidence of climate change in the region. The Standardized Precipitation Evapotranspiration Index (SPEI) at the monthly time scale was applied to detect and characterize wet and dry events for the period 1981-2014. The results of temporal variations of SPEI showed a strong change in basin climatic features associated with El Niño Southern Oscillation (ENSO) at the end of 1997, with the prevalence of wet and dry events before and after 1997 respectively. The results of spatial analysis show a higher susceptibility of the monsoon-dominated region towards wet events, with more intense events occurring in the eastern part, whereas a higher severity and duration is featuring in the southwestern part of the basin. In contrast, westerlies dominated region was found to be the hotspot of dry events with higher duration, severity, and intensity. Moreover, the surrounding region of the Himalaya divide line and the monsoon-dominated part of the basin were found to be the hotspots of rapid wet-dry transition events.

## 1. Introduction

There is growing evidence that recent warming is leading to significant alteration in hydrological cycle, exacerbating extreme weather events in general (Peterson et al., 2012) in many regions of the world. Extreme weather events such as floods and droughts and their rapid successions (recurrent spells) during past few decades have taken a heavy toll on both life and property. Moreover, such events can have large impacts on water availability, agriculture and food security, power production, and natural ecosystems (He et al., 2019, Sheffield and Wood, 2012). These events are projected to regionally intensify and be more frequent within the context of global warming, underscoring the importance of research on wet–dry extreme weather events collectively. The climate change projections for Asia continent in the sixth Assessment Report (AR6) of the Intergovernmental Panel on Climate Change (IPCC) reported that during the 21st century South Asia is likely to face more intense and frequent heatwaves and humid heat stress, whereas both annual and summer monsoon precipitation will increase, with enhanced inter-annual variability (medium confidence) (Zhongming et al. 2021). Various studies at local, basin, national and regional scale already documented and acknowledged the vulnerability to climate change of that region (He and Sheffield, 2020; Zhao et al. 2020; Visser-Quinn et al. 2019; He et al. 2017).

Typically, wet and dry events are generally considered independently in water resources management and planning. However, these events are inherently interconnected and governed by the same underlying hydrological processes and atmospheric dynamics, which may augment hydro-climatic variability under the influence of climate change (He and Sheffield, 2020). A number of wet-dry rapid altered events in the last decade

acknowledged the relevance of sequences of wet and dry events. For example, the California's large scale flood event in 2017 occurred at the offset of prolonged drought (2011-2016) (He et al., 2017, NOAA National Centers for Environmental Information, 2018). South Carolina observed an abrupt transition (within a week) from drought to flood in September 2015 (He and Sheffield, 2020). Other examples include the successive drought and flood events of 2010–2012 and 2015–2016 in the UK (Parry et al., 2013) and Tasmania, Australia respectively (CSIRO, 2018). Such abrupt flood-drought transitions put a substantial risk for water management practices, especially for reservoir operation, as a trade-off should be set between short-term flood-control and long-term water-storage imperatives to satisfy water demand (He and Sheffield, 2020). This has aroused widespread concern in the scientific community to understand the wet-dry interplay under a changing environment.

During the past few decades, significant effort was put forward towards the adoption of multi-hazard approach (consideration of both types of extreme hydrological conditions at the same time) in developing resilience to climate change. (Kourgialas, 2021) analyzed floods and droughts collectively in the Mediterranean agricultural region, and proposed water-saving and flood protection measures for adapting to the inevitable adverse effects of climate change. (Visser-Quinn et al., 2019) identified hotspots regions in UK where spatio-temporally concurrent increase in the number of flood and drought events were projected. (Zhao et al., 2020) investigated the rapid transition of flood and drought events under present and future climate change in the Hanjiang Basin and found more frequent drought to flood rapid transition events of higher intensity in the 21st century. Other examples include the analysis of rapid drought to flood transitions in river basins in China (Yan et al., 2013) and in England and Wales (Parry et al., 2013). These studies employed peak over threshold (POT) method and various indices recommended by the World Meteorological Organization (WMO) for the detection and characterization of wet-dry extreme events (floods and droughts).

Some commonly used indices are the Standardized Precipitation Index (SPI) (McKee et al., 1993), Standardized Precipitation Evapotranspiration Index (SPEI) (Vicente-Serrano et al., 2010), Palmer Drought Severity Index (PDSI) (Palmer, 1965), normalized difference vegetation index (NDVI) (Tucker, 1979), Standardized Drought Indices (SDI) (Svoboda and Fuchs, 2016), and Standardized Anomaly Index (SAI) (Katz and Glantz, 1986). Among these indices, SPI and SPEI are more widely accepted for the following reasons: a) simple to calculate, b) require few input data (precipitation and temperature), that are easily accessible in most cases, c) standardized indices, which facilitate the comparison of different climatic zones, and d) can be calculated at multiple timescales, depending on the objective. For instance, SPI and SPEI at short timescales (1, 2, 3 or 6-month) better reflect the meteorological and agricultural drought, while longer time scales (12, 24 or 48 months) are usually considered in hydrology (Kourgialas, 2021). The calculation of SPI and SPEI is mathematically similar, but it differs in the input parameters. The SPI index only uses precipitation, whereas SPEI is based on the climatic water balance. Many studies advocate the use of SPEI, rather than SPI, due to its link to potential evapotranspiration (PET), which makes it more sensitive in the context of global warming (Himayoun and Roshni, 2019, Yao et al., 2018, Huang et al., 2017, Vicente-Serrano et al., 2010).

In this study, attempts were made to understand the regional evolution of wet-dry events collectively, their characteristics and transition (wet-to-dry and dry-to-wet) for different severity levels ranging from moderate to extreme. Here, the term "wet and dry events" does not necessarily imply observed flood and drought events, unless explicitly mentioned. There exists a basic difference between a flood and a wet event. The former has a

short duration effect (e.g., a few hours or days) while the latter is regarded as a long period without precipitation

shortage (e.g., several months or years) (Wu and Chen, 2019).

The proposed framework was implemented with reference to the Upper Jhelum Basin (UJB), where the relevance of wet-dry events and their transition have not been assessed yet, despite of clear evidence of climate change in the region. The UJB is located in Western Himalaya and shared by Pakistan and India. The region already witnessed an increase in extreme hydro-meteorological events in the last few decades, but these events

are expected to become even more pronounced in the coming future (Pachauri et al., 2014). A study conducted over the Northern Highlands of Pakistan investigated the trends in time distribution patterns (TDPs) and return periods for event based extreme precipitation for a period of 1961 to 2014 and found maximum values of 20 and 50-year return levels of TDP for the UJB (Zaman et al. 2020). Another study conducted on a portion of UJB located in Kashmir, India, uses SPEI index for spatio-temporal characterization of drought events only (Himayoun

and Roshni, 2019). (Akhtar et al., 2020) investigated the correlation of meteorological and hydrological drought using SPEI and standardized streamflow index (SSI) over the Upper Indus Basin (UIB), including UJB. They validated the results with historically prolonged drought event observed in Pakistan (1999-02). Another study employed locally weighted SDI index and compared it with SPI and SPEI on ten meteorological stations within Pakistan (Ali et al., 2019). (Ullah et al., 2021a) evaluated four reanalysis products for drought assessment in

Pakistan using SPI and SPEI at multiple time scales. All above mentioned studies put focus towards drought event characteristics only, whereas the wet events and transition of wet-dry events were overlooked. This study attempts to fill this gap by addressing the following specific points:

1.  How does climate change influence the evolution of the regional wet-dry events?

2.  How comparatively frequent were wet or dry events in the past?

3.  What is the average transition time from wet-to-dry and dry-to-wet events?

4.  Which parts of the basin are hosting hotspots for rapid wet-dry transition events?

The most widely used index, SPEI, is here adopted to detect and characterize wet and dry events of different severity levels (moderate, severe, and extreme). The analysis was carried out both at each grid cell and averaged over the basin, using corrected ERA5 precipitation and observed temperature data for a period of 35 years (1981-

2014).

**2. Characterization of the study area**

The Upper Jhelum Basin (UJB) has a latitudinal extent stretching from 73° 07′ E to 75° 40′ E and latitudinal extent from 33° 00 ′ N to 35° 12′ N (Figure 1). The basin is mainly located in sub-tropics and partially in a temperate region. The basin drains the foothills of Western Himalaya and Pir-Panjal mountains and feeds the

second largest reservoir of Pakistan "Mangla Reservoir". The total area of the basin is about 33,342 km². The elevation ranges from nearly 223 m in the southwest to about 6201 m in the north, with mean elevation of 2353 m ASL. Approximately 0.75% (252 km²) of the basin is covered by perennial glaciers in the north of the basin (Consortium and Inventory, 2017). Grass, forest, and agriculture are the three major land use land cover (LULC) dominating over high, mid, and low elevation areas, respectively. Permanent snow and ice cover a negligible area

in the northwest of the basin whereas the small patch of barren land exists over the densely grassy mountains of western Himalaya and Pir-Panjal. The urban settlement covers a small portion of the basin, concentrated in the Kashmir valley.

The climate of the UJB is influenced by dynamic local and regional weather systems and the topography of the high mountains causes a huge variability in the spatial and seasonal distribution of precipitation (Dolk et al., 2020). Two distinct precipitation patterns (i.e., western disturbances and monsoon) exist in the basin. The western disturbances bring precipitation in the form of snow during winter season. The monsoon pattern brings liquid rainfall during summer seasons. The monsoon precipitation pattern dominates in the two lower sub basins, i.e Poonch and Kanshi, and progressively loses strength northward towards the foothills of Western Himalaya, where the influence of western disturbances is predominant (Neelam and Kunhar sub basins). The basin average annual precipitation and temperature is about 1150 mm/year and 13.2°C, respectively. Owing to the steep rugged mountainous topography of the basin and consequent short lag time, the flow level in the river and its tributaries rises abruptly during a rainfall event (Dar et al., 2019). Major extreme events witnessed by the basin are primarily led by vigorous interactions of moisture-laden monsoon circulation and southward penetrating mid-latitude westerly troughs into the Himalayan region (Vellore et al., 2016).

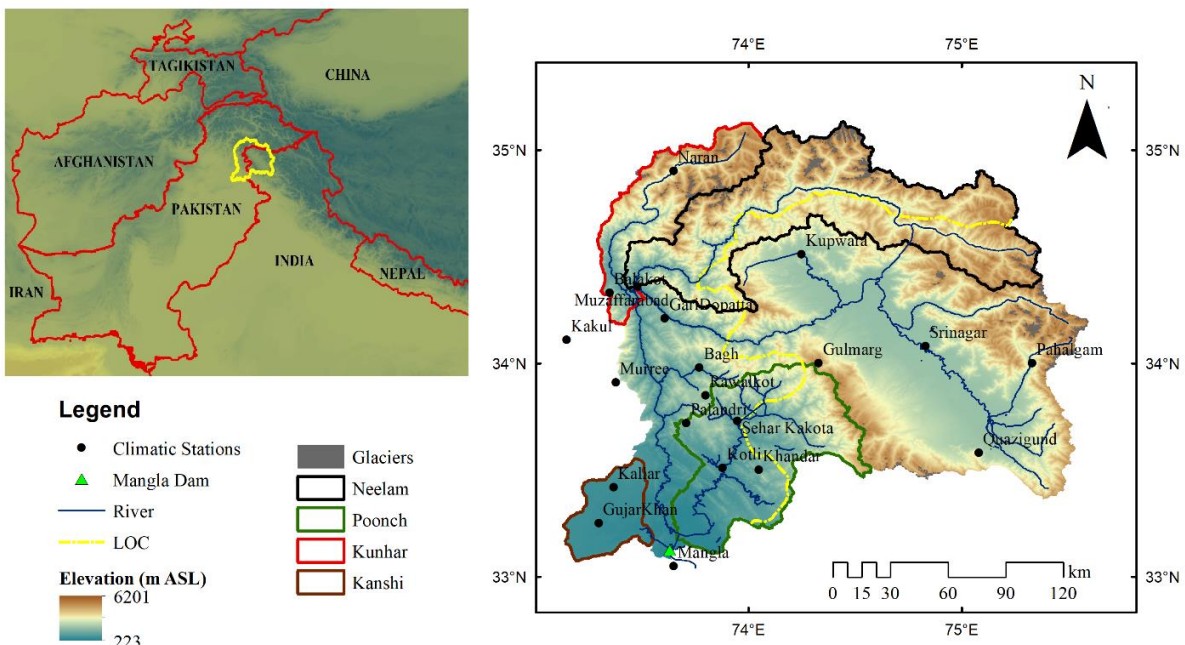

**Figure 1:** Location of the UJB and spatial distribution of climatic stations

### 3. Data description

The daily observed precipitation and temperature data of 15 climatic stations located within the political boundary of Pakistan were collected from Pakistan Meteorological Department (PMD) and Water and Power Development Authority (WAPDA). For the Indian side region, Indian Meteorological Department (IMD) daily gridded precipitation and temperature datasets, derived from a dense network of meteorological stations for the Indian mainland (Pai et al., 2015), were extracted at five stations and used for that region. The analysis was carried out for a period of 34 years (1981-2014), due to the availability of observed data. In fact there are only a few climatic stations where data are available starting from 1971, but the number of stations would not be enough for the spatial analysis. The observed temperature data was used to calculate potential evapotranspiration (PET) using the Thornthwaite equation (Thornthwaite, 1948) due to data limitation. A study conducted by Beguería et al.

(2014) compared the SPEI values calculated with three different methods (Penman-Monteith, Hargreaves, and

Thornthwaite) and found small differences in humid regions. Mavromatis (2007) also reported similar outcomes
of PET methods for drought indices calculation. Afterwards PET values were interpolated at $0.25^o$ using Kriging
with External Drift (KED), considering elevation as a predictor (Goovaerts, 2000). For the precipitation,
contrasting reviews are reported in the literature about the performance of KED technique. For instance, (Masson
et al., 2014) reported considerable improvement in interpolation accuracy with KED compared to other linear

regressions not accounting for any predictor in high mountainous regions. On the other hand, (Berndt and
Haberlandt, 2018, Ly et al., 2011) argue that topographical impact was indispensable for only temperature
reconstruction at all temporal resolutions and station densities, but its influence was less clear for daily to monthly
precipitation. Furthermore, all spatial interpolation techniques can perform poorly in regions with insufficient
high-elevation data, due to inaccurate estimation of local lapse rates (Ruelland and Sciences, 2020). Therefore,

the distribution mapping (DM)-corrected ERA5 precipitation estimates ($0.25^o$ horizontal resolution) were used in
the present study. ERA5 is a relatively new reanalysis launched by European Centre for Medium-Range Weather
Forecasts (ECMWF) (Saha et al., 2010). The data are developed by using advanced 4Dvar assimilation scheme
and provide various atmospheric variables at 139 pressure levels for the period 1979-present time. The suitability
of ERA5 to the UJB and surrounding region was also reported by Liaqat et al. (2021) and Baudouin et al. (2020).

The DM method adjusts the cumulative distribution function (CDF) of modelled precipitation to match with the
observed precipitation CDF using a transfer function (Sennikovs and Bethers, 2009) and it is commonly used to
correct the systematic distributional biases (Cannon et al., 2015). The Gamma distribution (Thom, 1958) with a
shape and a scale parameter was found to be suitable for the precipitation distribution in the study region (Azmat
et al., 2018). The suitability of ERA5 precipitation and bias correction method with respect to extreme

precipitation analysis was checked against observed station data and a few results of the reliability check of DM-
corrected ERA5 is provided in supplementary material (see figure S1).

## 4. Methods

### 4.1. Wet and Dry Events Identification

SPEI, a most widely used index was adopted to detect and characterize wet and dry events of different severity

levels (moderate, severe, and extreme). The SPEI index support comparisons over time and space, as proxies of
wet and dry conditions from both the meteorological and agricultural perspectives. Although the SPEI was
originally proposed for drought monitoring, it can also be used as a tool to detect flood risk. The calculation
procedure of SPEI involves two steps: fitting a log-logistic distribution to the monthly climatic water balance (P-
PET) time series and then transforming the cumulative probability of the fitted distribution to a standard normal

distribution (with mean zero and variance one). According to this distribution method, the probability distribution
function of a variable x is expressed as:

$$F(x) = [1 + (\frac{\alpha}{x-\gamma})^\beta]^{-1} \qquad (1)$$

Where α, β and γ are the shape, scale, and origin parameters, respectively. In the second step, SPEI is calculated
as the standardized value of F(x) as follows:

$$SPEI = W - \frac{C_o + C_1 W + C_2 W^2}{1 + d_1 W + d_2 W^2 + d_3 W^3} \qquad (2)$$

Where

$$W = \sqrt{-2\ln\big(F(x)\big)} \qquad \text{for F(x)} < 0.5 \tag{3}$$

$$W = \sqrt{-2\ln\big(1 - F(x)\big)} \qquad \text{for F(x)} > 0.5 \tag{4}$$

The parameters C0, C1, C2, d1, d2, d3 are SPEI constants (Vicente-serrano et al., 2010). The log-logistic
distribution for SPEI calculation was used and recommended by many researchers (Ullah et al., 2021a, Akhtar et
al., 2020, Himayoun and Roshni, 2019, Vicente-Serrano et al., 2010). The detailed description of the SPEI
calculation procedure can be found in (Vicente-Serrano et al., 2010). In this study, SPEI was calculated using
"SPEI" package in R environment (Beguería et al., 2017). The severity levels of wet and dry events based on
SPEI values were classified according to (Chen et al., 2020), results are listed in Table 1. Positive and negative
value of SPEI represent the severity of wet and dry events, respectively.

**Table 1**: SPEI Classification of Dry and Wet Events (from Chen et al., 2020)

| SPEI value | Description |
|---|---|
| > 1.99 | Extreme Wet |
| 1.99 to 1.50 | Severe Wet |
| 1.49 to 1.00 | Moderate Wet |
| 0.99 to -0.99 | Normal |
| -1.00 to -1.49 | Moderate Dry |
| -1.50 to -2.00 | Severe Dry |
| -2.00 < | Extreme Dry |

### 4.2. Wet and Dry Events Characteristics

In this study, three characteristics (severity, duration, and intensity) of wet and dry events were calculated
for each pixel. Following (Spinoni et al., 2014), the duration (D) of a wet/dry event is the length of time (months)
that the index is consecutively above or below a truncation value; the Severity (S) refers to the cumulative value
of the index from the first month to the last month of the wet/dry event and it represents the water surplus and
deficit, respectively ; the intensity (I) of an event is the ratio of severity (S) to duration (D). These characteristics
were computed for each event and then further the total wet/dry event duration (TWD and TDD), total wet/dry
severity (TWS and TDS), total wet/dry intensity (TWI and TDI), average wet/dry event duration (AWD and
ADD), average wet/dry severity (AWS and ADS), average wet/dry  intensity (AWI and ADI), maximum wet/dry
event duration (MWD and MDD), maximum wet/dry severity (MWS and MDS), maximum wet/dry  intensity
(MWI and MDI) were calculated for a period of 34 years (1981-2014).

### 4.3. Wet–Dry (WD) ratio

Wet-Dry (WD) ratio is defined as the natural logarithm of the ratio of the total number of wet months
($N_w$) to the total number of dry months ($N_d$) (Luca et al., 2020). The WD ratio was calculated for different levels
of severity (moderate, severe, and extreme) at each pixel for the studied period (1981–2014) using Eq. (5):

$$WD\ ratio = \ln\left(\frac{N_w}{N_d}\right) \tag{5}$$

The WD ratio provides information about the susceptibility of a given area to be more affected by wet or dry events. A WD ratio greater than 0 implies the prevalence of wet events whereas a WD ratio lower than 0 shows a dominance of dry events. The natural logarithm was used to narrow the range of WD ratio values and to separate the wet-dominated versus dry-dominated regions by sign.

### 4.4. Wet-Dry Transition Time

The total number of transitions and their average transition time ($T_t$) in months for wet-to-dry and dry-to-wet events was computed for each grid cell for the period 1981–2014, as described by (Luca et al., 2020). The calculation procedure of wet-to-dry transitions time ($T_t$) involves four steps: (i) extraction of wet and dry events and arrange them in an ascending order of time (from the oldest to the most recent); (ii) in case of consecutive dry and wet months, keep only the first and the last month value, respectively; (iii) calculate the difference in months between wet to dry events within the time series; and (iv) take the average of the time interval. The same procedure was applied for calculating dry-to-wet transitions time ($T_t$), with the only difference being in step (ii) in which the first and last month of wet and dry event were kept, respectively and in step (iii) in which the time interval was calculated between dry to wet events. The wet-to-dry and dry-to-wet transition time were calculated separately for each level of severity (moderate, severe, extreme).

### 4.5. Wet-Dry Rapid Transition Events

The wet-dry rapid transition event is defined as the consecutive occurrence of wet and dry months/events. For instance, a dry (or wet) event occurring in the ith month abruptly altered to wet (or dry) event in the i + 1st month. In this study, the frequency of wet-to-dry (wet event followed by dry event) and dry-to-wet (dry event followed by wet event) rapid transition events were calculated for each pixel to identify the geographical hotspot for compound extreme events. Unlike the wet/dry average transition time which were calculated separately for each severity level, the wet/dry rapid transition events were calculated considering all levels of severity together.

### 5. Results

#### 5.1. Change trends of the Wet-Dry Events

The basin average SPEI time series at 1-month (SPEI-1), 3-month (SPEI-3), 6-month (SPEI-6) and 12-month (SPEI-12) time scale is presented in figure 2. It can be seen that the study domain mostly experienced moderate-to-severe wet/dry events, whereas the extreme wet/dry events (SPEI>2 or SPEI<-2) rarely occurred during the study period. For the SPEI-1, the wet (blue) and dry (red) events changed more frequently than accumulated SPEI (at 3-, 6- and 12-month) and there was no extended dry or wet period. The reason might be that the precipitation and temperature of each new month has a substantial impact on the accumulative values of that period. By contrast, with the increase in SPEI time scale (SPEI-1 to SPEI-12), a clear change/shift of basin climate from wet to dry conditions can be seen (Figure 2), showing the stability in the frequency of incidence of wet/dry events over the study domain. This could be explained as the slow and consistent response of SPEI towards changes in climatic variables, indicating strong and clear durations of annual and multiple-year dry and wet conditions. This means that at longer time scales of SPEI the number of occurrences of wet/dry events will decrease, but the duration will increase.

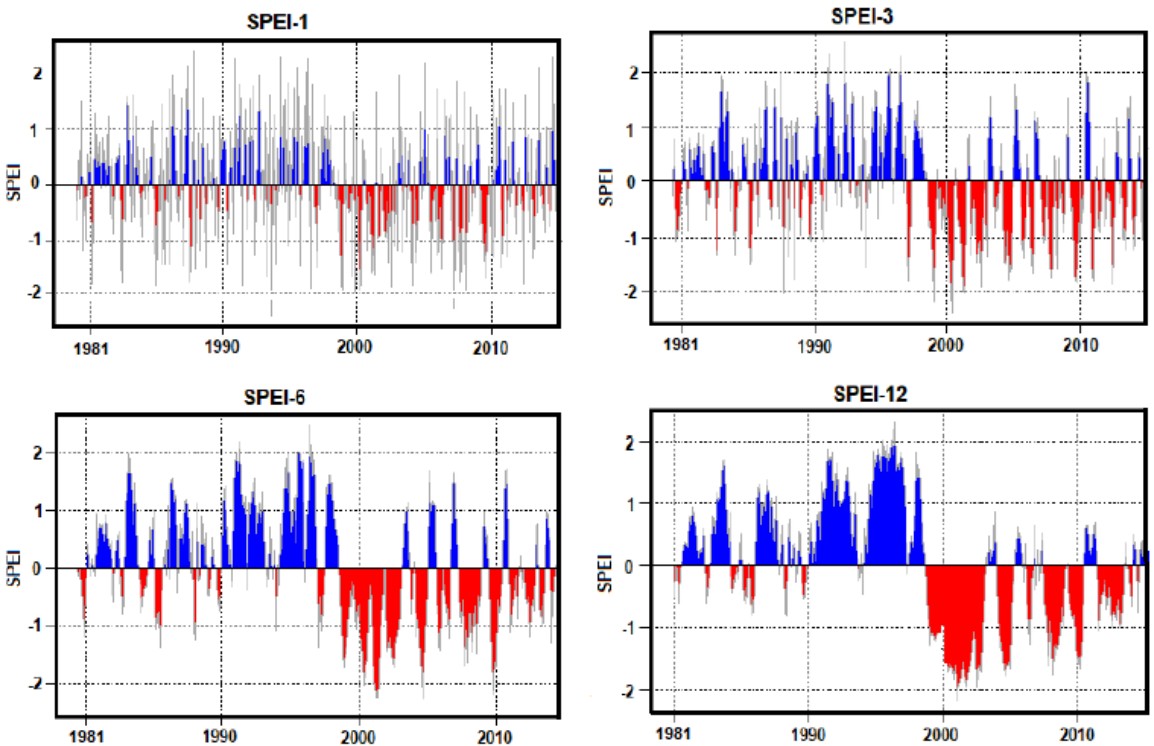

**Figure 2:** Temporal variations of SPEI at 1-, 3-, 6- and 12-month time scale over UJB for the period 1981-2014

This study focuses on the short time scale conditions to analyze frequent variations in climatic conditions and their interplay; therefore, more detailed analysis was carried out at the monthly time scale. Moreover, the floods and flash droughts are not clearly associated with long term SPEI, because the averaging effect of long-term accumulated precipitation and temperature surpasses the signal of extreme precipitation and temperature over a short period. Flash drought is relatively a new type of drought. Currently, there is not a universally accepted definition or criteria for flash drought, though there is general consensus on the principle of rapid onset or intensification characterized by moisture deficits and abnormally high temperatures for a period lasting at least 3 weeks (Lisonbee et al. 2021, Otkin et al. 2018, Hunt et al. 2009). This highlights the usefulness of SPEI at the monthly scale in representing flood and flash drought events. It is noted that the terms "wet-dry events" or "wet-dry months" present similar meaning for our study, as the analysis was made at the monthly time step. A clearer picture of the monthly evolution of wet/dry events of different severity levels and their variability can be seen in Table 2. The SPEI-1 values fluctuate remarkably from one month to another. For example, an extreme wet October in 1987 was followed by a severe dry November, and a severe wet June occurred at the tail of the longest drought spell in May 2001. Such rapid transition from wet to dry and from dry to wet events was more prominent during the first half of the study period (before the year 1997). Another interesting observation concerns the strong change in the basin climatic features which can be noticed around the year 1997/1998. During the first half of the study period (1981-1997), the dominancy of wet events of different categories prevails whereas the basin conditions lean towards dryer conditions during the second half of the period (1998-2014).

**Table 2**: Temporal variations of monthly SPEI over UJB from 1981-2014. The brown, blue and white colors present dry, wet and normal months, respectively. Different shades of the colors define the different severity

levels (EW-extreme wet, ED-extreme dry, SW-severe wet, SD-severe dry, MW-moderate wet, MD-moderate dry). The red line between 1997 and 1998 indicates the strong change in the basin climatic features

| Year/Months | 1 | 2 | 3 | 4 | 5 | 6 | 7 | 8 | 9 | 10 | 11 | 12 |
|---|---|---|---|---|---|---|---|---|---|---|---|---|
| 1981 | | | | | | | MW | | | | | |
| 1982 | | | | | | MD | SD | | | MW | SW | MW |
| 1983 | | | SW | MW | | | | MW | | | | |
| 1984 | | | | | | | | | MW | MD | | |
| 1985 | | SD | | | | MD | MW | | MD | | | SW |
| 1986 | MD | | SW | MW | | | | | | | SW | MW |
| 1987 | MD | | | MW | EW | | SD | MD | | EW | SD | |
| 1988 | | | MW | MD | MD | | SW | | | | MD | |
| 1989 | | | | | MW | | | | | | MW | |
| 1990 | | | | | MD | | | | | | | EW |
| 1991 | | | | SW | MW | SD | | | MW | | | |
| 1992 | SW | | SW | | | MD | | | SW | | | |
| 1993 | | | MW | SD | | | SW | ED | | | MW | MD |
| 1994 | | | | EW | | | MW | SW | | | MD | SW |
| 1995 | | | | SW | | | EW | MW | MD | | | |
| 1996 | | | MW | | SW | EW | | MW | SD | SW | | |
| 1997 | | MD | | | MW | MW | | SW | | MW | | |
| 1998 | | MW | | | | | | | | | SD | SD |
| 1999 | MW | MD | | | | | | | | SD | MW | SD |
| 2000 | | MD | MD | SD | SD | | | | | MD | | |
| 2001 | SD | SD | MD | | SD | SW | MW | | | | | |
| 2002 | | | | | MD | | SD | | | | MD | |
| 2003 | MD | SW | | | | MD | | | | | | |
| 2004 | | MD | SD | | MD | | MD | | | MW | | |
| 2005 | | EW | | | | | | SD | | | | MD |
| 2006 | SW | | MD | | SD | | | SW | | | MW | MW |
| 2007 | SD | | | ED | | SW | | MD | | SD | MD | |
| 2008 | SW | | SD | | | MW | | | | | | MW |
| 2009 | | | | | | | MD | | SD | | | |
| 2010 | MD | SW | MD | | MW | | SW | MW | | | MD | MD |
| 2011 | | SW | | | MD | | | | MW | | | |
| 2012 | | | | | | SD | SD | | SW | | | |
| 2013 | | | MD | MD | | | | EW | | | | |
| 2014 | | | MW | | | | | | EW | | | MD |

Annual variations in the number of months affected by dry/wet events (SPEI $\leq$ -1 and SPEI $\geq$ 1) is displayed in figure 3. Usually, every year encountered at least one dry and wet month of any severity level. Approximately 35% of the total number of months experienced anomalous dry or wet conditions. The proportion of wet months (18.1%) was slightly higher than that of dry ones (16.9%).

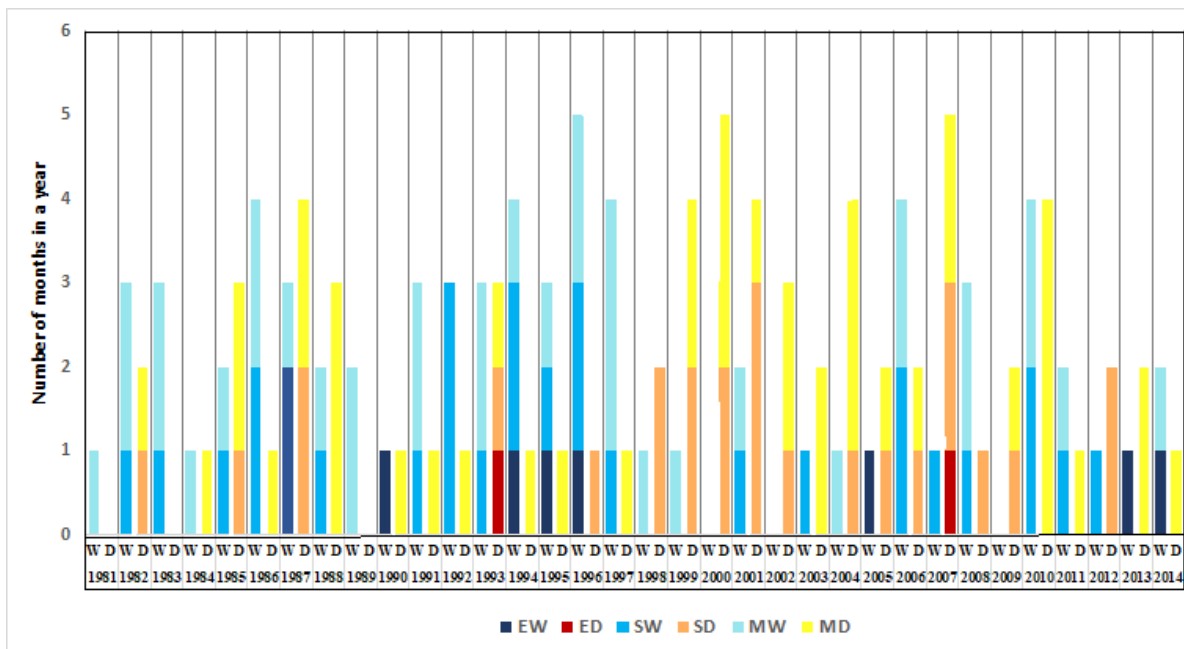

**Figure 3:** Annual variations in the number of months affected by wet/dry conditions during 1984-2014. The brown and blue colors present dry and wet months, respectively. Different shades of the colors define the different severity levels (EW-extreme wet, ED-extreme dry, SW-severe wet, SD-severe dry, MW-moderate wet, MD-moderate dry)

### 5.2. Wet/Dry Event Analysis

The wet/dry event characteristics (duration, severity, and intensity) were computed for each pixel to analyze their spatial distribution. Pixel based analysis shows the location of the most vulnerable parts of the basin, providing the basis for future decisions on adaptation and mitigation measures. In this study, the total, average and maximum value of duration, severity, and intensity were computed for the study period (1981-2014). The maps of wet and dry duration are displayed in figure 4. Overall, the study area encountered relatively more wet months than dry months during the whole study period. The total wet duration (TWD) and the total dry duration (TDD) vary from 66 to 80 and from 61 to 65 months respectively for most part of the basin. The low elevation parts in the south of the basin show the highest value of TWD whereas the TDD is higher across Himalaya divide line than in other parts of the basin. The Himalayas divide line is a line in the middle of the UJB at Pir Panjal mountainous range, separating the dominance of the two precipitation patterns: westerlies in the north-facing and monsoon in the south-facing slopes of the line (Archer and Fowler, 2008).

The average wet and dry event durations (AWD & ADD) were found to be similar throughout the basin with a slight difference in the range of 1-2 weeks. However, their spatial patterns were found to be mostly complementary. Maximum wet and dry event durations (MWD & MDD) exhibit high values in two distinct parts of the basin. The MWD is about 6-7 months in the east of the basin, which is located in Kashmir, India, whereas it varies between about 4-5 months and 2-3 months in the northwest and southwest parts of the basin. For the MDD, the northwest and central parts of the basin show higher values (4-5 months) than the remaining parts (2-3 months).

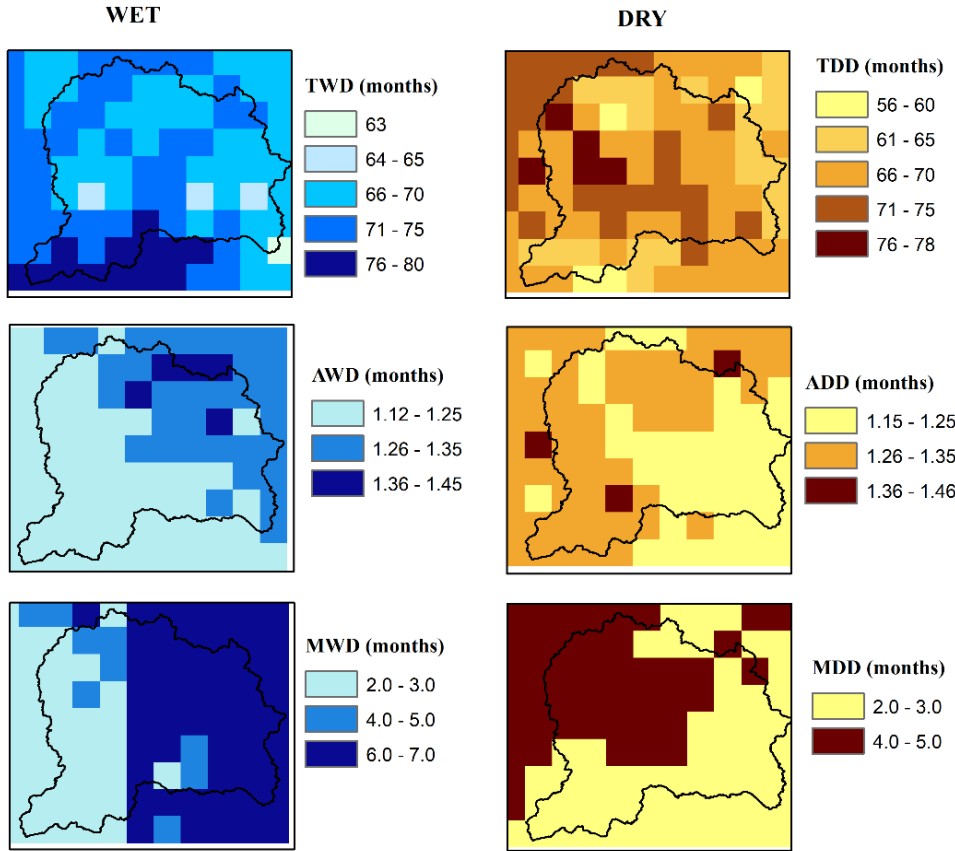

**Figure 4:** Spatial distribution of total wet duration (TWD), total dry duration (TDD), average wet duration (AWD), average dry duration (ADD), maximum wet duration (MWD) and maximum dry duration (MDD) for the period 1981-2014

The spatial distribution of total, average and maximum severity of wet/ dry events is presented in figure 5. All wet/ dry severity maps show similar spatial patterns as wet/dry duration maps. In terms of total wet severity (TWS) and total dry severity (TDS), the wet and dry hotspots are located in the south and middle (across Himalaya divide line) of the basin, respectively. Unlike the spatial patterns of TDD, the TDS is relatively higher in the north of the basin above Himalaya divide line. This shows more intense dry events in this part of the basin. The underlying reason for higher TDS could be the higher warming rates in western Himalaya, hosted in the north of the basin. The average severity of wet and dry events is categorized from moderate to severe level. The average wet severity (AWS) exhibits random spatial patterns, whereas the average dry severity (ADS) is relatively higher in the north of the basin. Observed spatial patterns of maximum wet severity (MWS) and maximum dry severity (MDS) were similar to those of MWD and MDD. The eastern part of the basin experienced wet events of higher severity than the western one, whereas the most severe dry events affected the northwest and central parts of the basin.

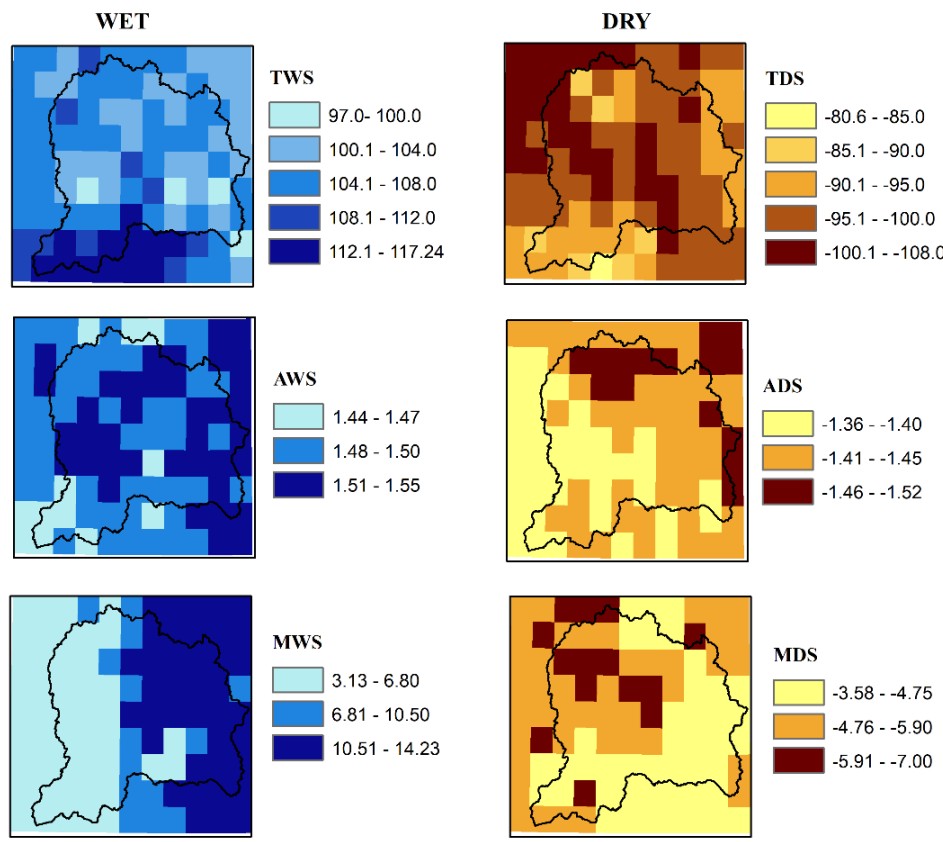

**Figure 5:** Spatial distribution of total wet severity (TWS), total dry severity (TDS), average wet severity (AWS), average dry severity (ADS), maximum wet severity (MWS) and maximum dry severity (MDS) for the period 1981-2014

Figures 6 illustrates the spatial distribution of intensities of wet/dry events, calculated as the ratio of
325 severity to duration. The total wet intensity (TWI) and total dry intensity (TDI) varies from moderate to severe with noted range of 1.44 to 1.55 and -1.36 to -1.52 for wet and dry events, respectively. Irrespective to TWD and TWS, which is highest in the south of the basin, TWI is more intense in the middle and northeast of the basin. The TDI is found to be more intense over western Himalaya Mountains, north of the basin. The average wet intensity (AWI) and average dry intensity (ADI) vary within the moderate class of hazard. However, their spatial patterns
are much different from average duration (AWD & ADD) and average severity (AWS & ADS) patterns. Regarding maximum intensities, the spatial patterns of maximum wet intensity (MWI) well resemble the patterns of MWD and MWS, whereas the maximum dry intensity (MDI) exhibits much different spatial patterns from MDD and MDS. The dry events are found to be more intense than wet events, but only for a few pixels in the southwest of the basin. On the other hand, wet events with higher intensities are found to be more widespread
than dry events.

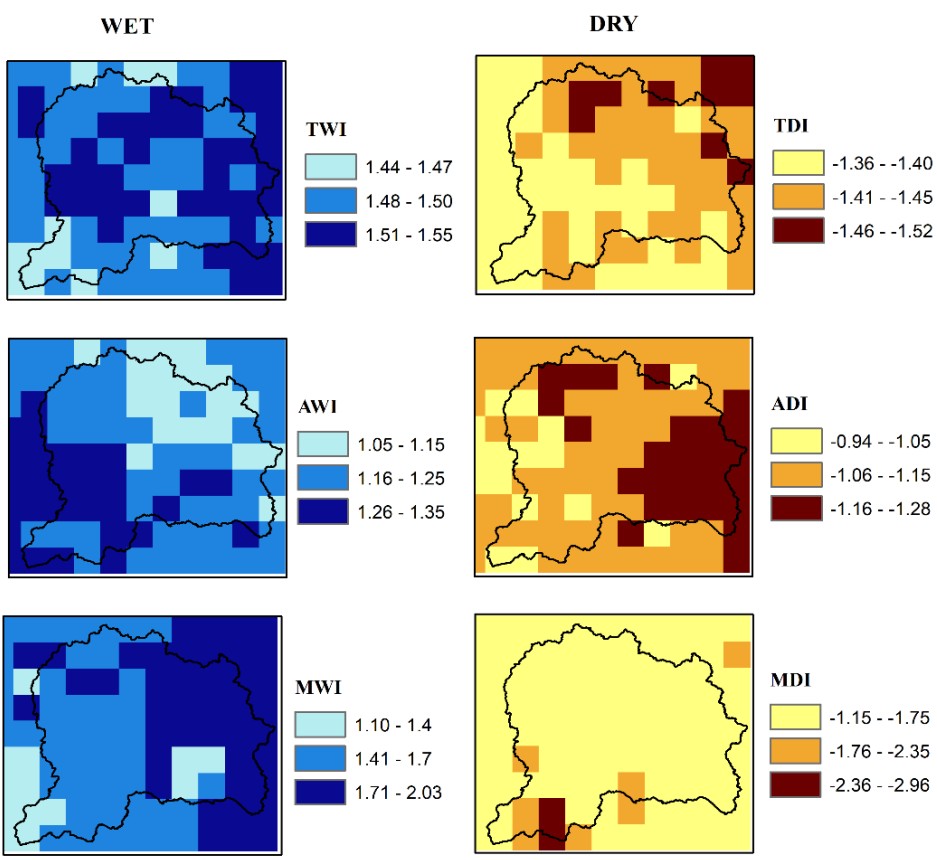

**Figure 6:** Spatial distribution of total wet intensity (TWI), total dry intensity (TDI), average wet intensity (AWI), average dry intensity (ADI), maximum wet intensity (MWI) and maximum dry intensity (MDI) for the period 1981-2014

### 5.3. Wet-Dry Ratio

The WD ratio features the dominance of wet or dry events for the period of 34 years (1981-2014). The WD ratio for the three severity levels (moderate, severe, and extreme) at pixel basis is presented in figure 7. The positive and negative value of WD ratio depicts the prevalence of wet and dry events, respectively. As the figure shows, higher frequencies of moderate dry events with respect to moderate wet events were found throughout the basin except a few pixels in the south. By contrast, severe to extreme wet events are more frequent for most parts of the basin. The highest positive values of WD ratio for extreme level of hazard were found in the southwest of the basin, which shows the higher susceptibility of the area towards extreme wet events. Moreover, the analysis of wet/dry event characteristics also revealed the prevalence of wet events with higher duration and severity over monsoon dominated region.

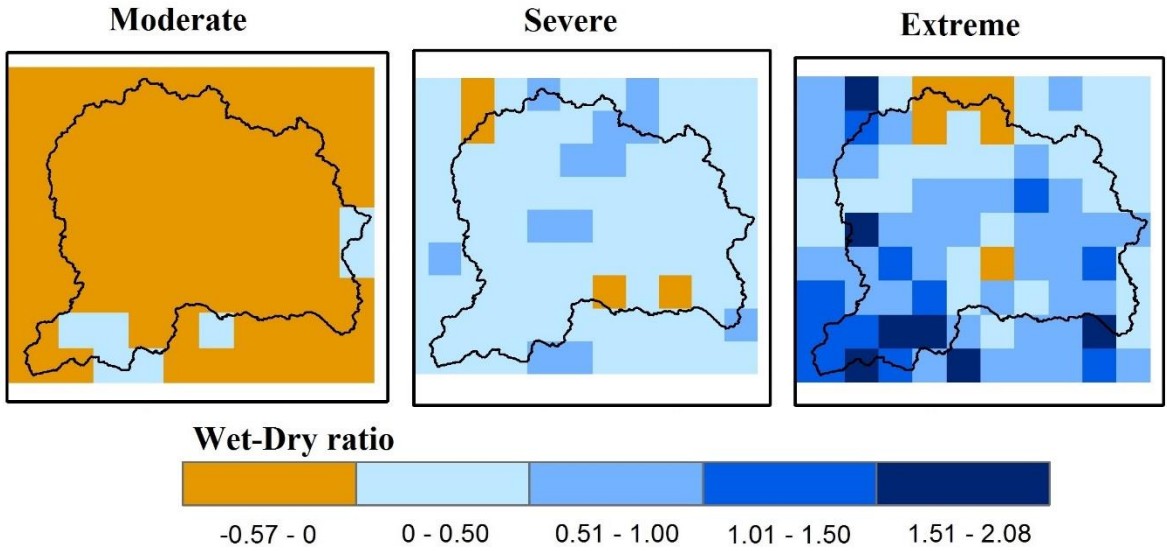

**Figure 7:** Spatial distribution of wet–dry (WD) ratio derived for three levels of severity (moderate, severe and extreme) during the period 1981-2014. Blue (WD ratio>0) means that the area experienced more wet than dry events. Brown (WD ratio<0) indicates the opposite.

### 5.4. Wet-Dry Transition Time

The number of transitions and their average transition time for wet-to-dry and dry-to-wet events for the period 1981–2014 is presented in figure 8 and figure 9. As expected, the number of transitions for wet-to-dry and dry-to-wet event was the highest for the moderate level of events, followed by severe and extreme levels of events. Consequently the average transition time from wet-to-dry and dry-to-wet event was found to be the highest for the extreme level of event followed by severe and moderate levels of events. The number of transitions for moderate, severe, and extreme levels of events varies from 15 to 26, from 6 to 16, and from 1 to 5 respectively. Overall, the number of transitions for dry-to-wet event is larger than the wet-to-dry event for severe and extreme levels of events, whereas the opposite was found for the moderate level of events. The transition time for moderate, severe, and extreme levels of events varies from 1.8 to 6.5, from 1.8 to 16.75 and from 3.5 to 187.0 months, respectively. Overall, 53.57% and 17.86% of pixels in the UJB showed longer transition time from wet-to-dry than from dry-to-wet for moderate and extreme levels, whereas the opposite was seen for severe events.

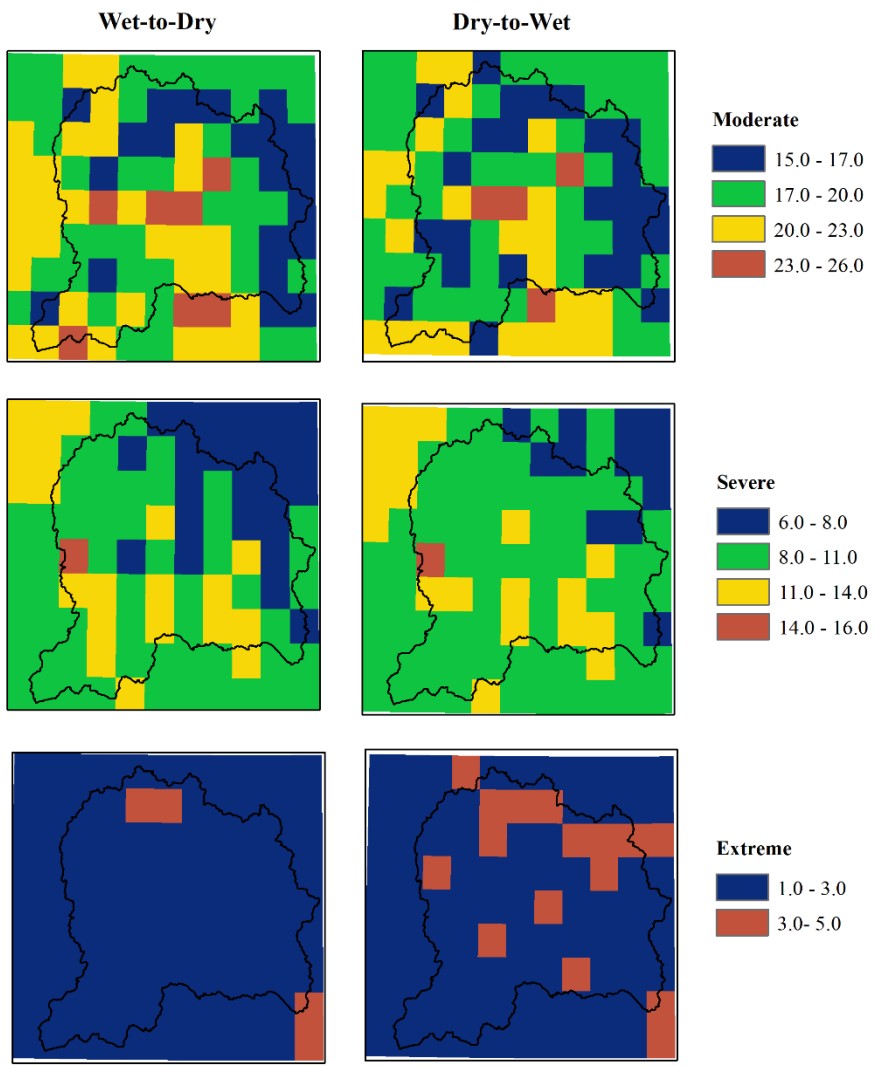

**Figure 8:** Number of transitions from wet-to-dry (left) and dry-to-wet (right) events for three levels of severity (moderate, severe, extreme) for the period 1981-2014

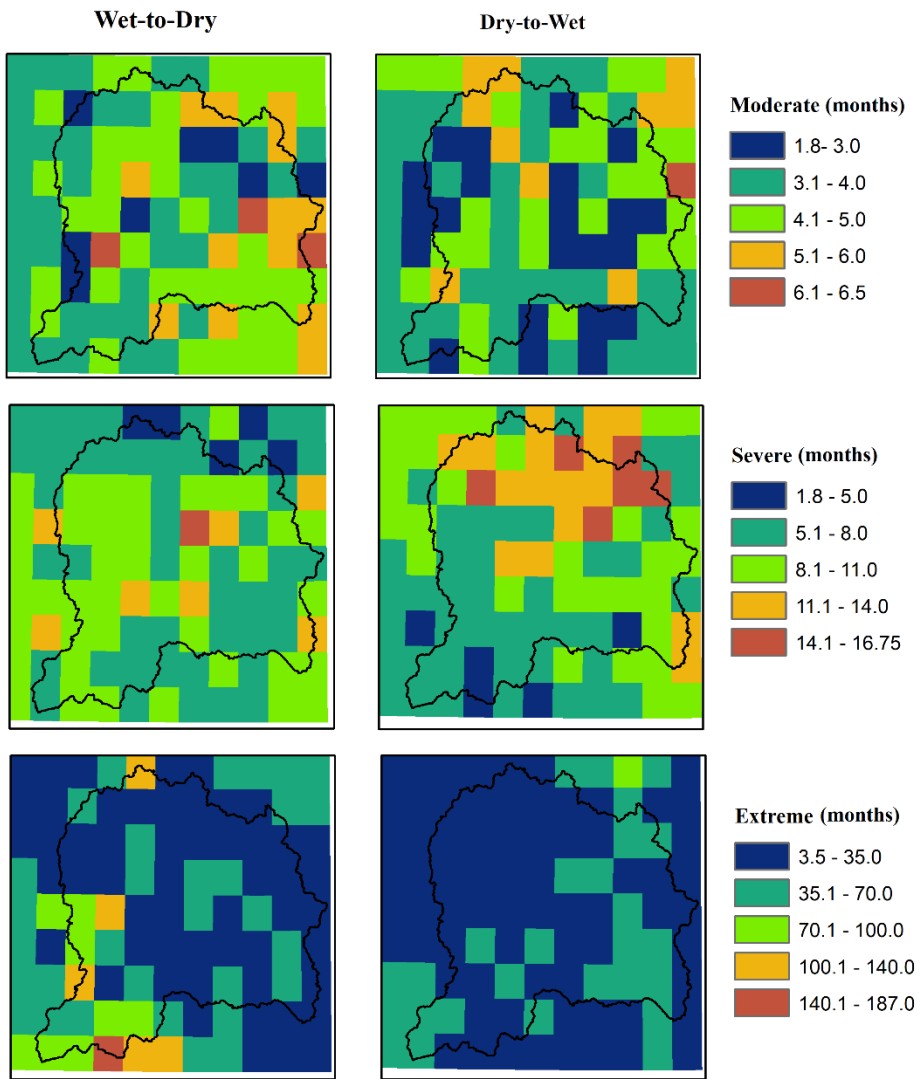

**Figure 9:** Average transition time (Tt) intervals in months for wet-to-dry (left) and dry-to-wet (right) events for three levels of severity (moderate, severe, extreme) for the period 1981-2014

### 5.5. Wet/Dry Rapid Transition Events

The wet/dry rapid transition is the consecutive occurrence of wet and dry months of any severity level. The frequency of wet-to-dry (wet month followed by dry month) and dry-to-wet (dry month followed by wet month) rapid transition events were computed for each grid cell and are shown in figure 10. The frequency of wet/dry transition events varies/ranges from 5 to 20 events during 34 years of study period. About 50% pixels in the UJB encountered more number of wet events terminated at dry months. The spatial distribution of frequency of wet/dry rapid transition events revealed that the wet-to-dry events are less frequent over the westerlies dominated region of the basin, whereas the southwestern part of the basin was more affected by the wet-to-dry abrupt altered events. By contrast, dry-to-wet abrupt altered events are found to be more frequent over pixels surrounding the Himalaya divide line, whereas the remaining part of the basin depicts less incidence of dry-to-wet altered events.

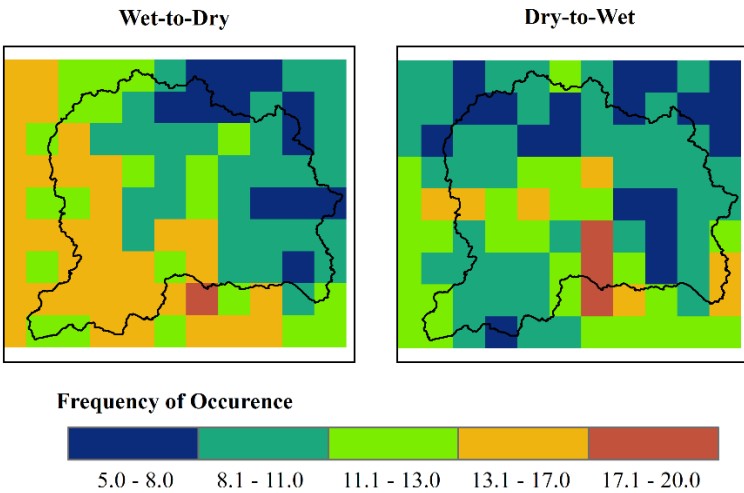

**Figure 10:** Frequency of occurrence of abrupt altered events, wet-to-dry (left) and dry-to-wet (right) during the period 1981-2014

## 6.  Discussion and Conclusion

This study attempts to investigate the spatiotemporal variations of wet-dry events collectively, their characteristics (duration, severity, intensity) and transition from wet-to-dry and dry-to-wet events during the period 1981-2014 in the Upper Jhelum Basin (UJB)-South Asia. The SPEI index, which incorporates both precipitation and potential evapotranspiration, was used to extract and analyze the wet-dry events. The whole analysis was carried out at the monthly time scale, but the temporal evolution of the basin averaged index was also simulated at multiple time scales (1-, 3-, 6- and 12-months). The reason for selecting the monthly time scale for this study is that it is expected to provide the best performance in detecting floods and flash droughts, as longer time steps are more appropriate for long term droughts only and not for floods.

The results of temporal variations of SPEI showed that the study domain mostly encountered moderate to severe wet-dry events, whereas the extreme wet-dry events rarely occurred during the study period. The results of basin average SPEI at multiple time scales revealed that the response of SPEI to the deviations in climatic features varies with the accumulation time. Therefore, shorter time scales are more appropriate for detecting frequent seasonal and inter-annual variations, whereas longer time scales provide useful information regarding the signature of the events over the region (Ayugi et al., 2020, Du et al., 2013). Furthermore, the SPEI time-series plots well capture the observed extreme floods and drought events occurred in the basin during the study period: for instance, the longest drought event occurred from late 1990s to early 2000s, as evident in Figure 2 and Table 2. The drought started in 1998 and was considered to be the worst in the history of Pakistan. The drought spell 2001-2002 resulted in water shortage of up to 51% of normal supplies (Ahmad et al., 2004). Likewise, the notable flooding events, usually flash floods ranging from moderate to severe, occurred in the years 1988, 1992, 1994, 1997, 2007 and 2014 (Bhat et al., 2019) and were well captured by SPEI index, confirming its valuable contribution to this type of analysis.

An interesting clue to the changing climate is the strong change occurred in the basin at the end of 1997 (Table 2). Before this change (1981-1997), wet events of different severity levels predominated in the basin, whereas dryer conditions prevailed after 1997. However, it still needs to be investigated whether dryer conditions are expected to continue in the future, or a large multi-decadal variation is taking place. This strong change in the

basin climate coincides with the strongest El Niño Southern Oscillation (ENSO) event in the winter season of 1997-1998, where the Oceanic Niño Index (ONI) peaked at 2.3, and influenced the climate conditions all over the world (MRCC, 2021). The 1998-2002 drought in southwestern Asia, accompanied by the most severe drought conditions in the last 50 years, was also a result of this strong ENSO event (Ain et al., 2020, Ahmed et al., 2018). The ENSO is the primary mode of inter-annual variability having great influence on global weather and climate via atmospheric circulations (Ullah et al., 2021a). Many researchers reported the close association between variations in atmospheric circulation patterns and climatic variables, extreme weather phenomena like drought and flood (Luca et al., 2020, Omidvar et al., 2016, Sun et al., 2015). (Kenyon and Hegerl, 2010) examined the response patterns of hydroclimate extremes to ENSO over global land areas, and stated a significant decrease in precipitation extremes over Southeast Asia, Indonesia, Australia, and the northernmost region of South America during El Niño phases, whereas in the southern tier of the United States and the region from Argentina to southern Brazil heavy precipitation increased during El Niño phases, and vice versa during La Niña phases. The strength of such connections for Pakistan was also demonstrated in several studies. El Nino suppresses monsoon rainfall activity over Pakistan, while La Nina has a negative impact on winter precipitation over Pakistan (Farooqi et al., 2005, Azmat, 2003).(Ullah et al., 2021a) found significant impacts of three large scale climate indices, i.e Niño4-SST Index, Sea Surface Temperature (SST), and multivariate El Niño-Southern Oscillation (ENSO4.0) on seasonal droughts across Pakistan.

The results of wet-dry event characteristics (duration, severity, intensity) at pixel basis outline the greater susceptibility of westerlies dominated region towards dry events with higher duration, severity, and intensity. The dryer conditions in this region could be explained with the increasing rates of global warming over the mountainous region of the basin, also reported by many researchers (Rashid et al., 2020, Shafiq et al., 2020, Zaz et al., 2019). Studies by (Negi et al., 2018) and (Dimri and Dash, 2012) also confirm that most of the western Himalayan region recorded a significant warming trend especially from 1975 onwards. This is also supported by the tree-ring chronologies of the region which indicate a rapid growth of the tree rings in the recent decades especially at higher altitudes (Borgaonkar et al., 2009). The impact of global warming on short term dry event (soil moisture drought) is not straightforward as rising temperature did not necessarily cause increase in actual ET, especially in arid and semiarid regions (Trenberth et al., 2014, Sheffield et al., 2012). In fact the rate and amount of ET results from a complex interaction of temperature, radiation balance, precipitation rates and vegetation physiological control, rather than being exclusively limited by one of these factors. For flash drought, the rapid soil moisture decline should be a result of the intensification of ET driven by higher temperature, which is very common in humid and semi-humid regions, where soil moisture can sustain higher ET amounts up to a few weeks (Yuan et al., 2019). Further decrease in winter and spring precipitation leads water deficit conditions in this part of the basin. The worst drought event period (2000-2001), partially induced by a stronger ENSO in winter, was also due to the low winter and spring precipitation, as shown in Table 2. During 2000-2001, winter and spring seasons were moderate-to-severe dry, whereas the monsoon and autumn seasons observed normal months. By contrast, the higher duration and severity of wet events were detected in the monsoon dominated region, implying that floods mainly occurred during monsoon season with heavy rainfall along with snowmelt. However, the eastern part of the basin was the hotspot of more intense wet events. The above discussion is also supported by the historic database of observed flood events, as most of these events occurred during monsoon season.

The results of WD ratio showed the prevalence of severe to extreme wet events for most part of the basin, while the dry events of moderate severity level were more frequent in the study domain. The southwestern part of the basin, located in the monsoon-dominated region was found to be the hotspot for the extreme wet events. Moreover, the analysis of wet/dry event characteristics also revealed the prevalence of wet events with higher duration and severity over the same monsoon dominated region. The spatial patterns of average transition time from one extreme type to the other type was found to be heterogeneous and different for the three severity levels. Overall, a greater number of pixels took shorter time to switch from dry to wet event than from wet to dry events. Apart from the average transition period, the study domain also experienced rapid transition of wet-dry events. In general, the surrounding region of the Himalaya divide line and the monsoon-dominated part of the basin were found to be the hotspots of rapid wet-dry transition. The rapid wet-dry swings could be explained in the context of global warming. In a warmer climate, increased evapotranspiration rates in response to increased temperature could elevate the drought risk and frequency. At the same time, prospect of localized heavy precipitation causing floods is expected to increase in response to increased atmospheric moisture content due to increased evapotranspiration rates (He and Sheffield, 2020, Krishnan et al., 2020). Further warming-induced changes in global climate variability, such as El Niño and La Niña can cause more inter-annual variability or persistence in global weather and climate, significantly affecting regional precipitation and temperature distribution in space and time (Ullah et al., 2021b). Further compelling scientific evidence of human interventions, such as boosted human water intake and land use changes, exacerbate the extreme flood and drought risk hazard.

To conclude, knowledge of wet-dry events characteristics and their rapid transition provides meaningful insight into the geographical hotspots of compound extreme events, which could be of practical value to inform a group of stakeholders (researchers, local authorities, policy makers, relief agencies, non-governmental organizations (NGOs) and (re)insurance companies) on the potential risk. In general, results contribute to hydrological predictability and risk assessment and therefore effectively support disaster preparedness and risk management, ensuring the regional water, food and socio-economic security and stability against the background of a changing environment. Future work should explore to what extent future wet-dry event frequency will respond to anthropogenic forcing, internal atmospheric processes, and human interventions.

**Author contribution:** This paper was conceptualized by RA and GG. RA performed the data analysis and visualization. The original draft was written by RA and revised by GG.

**Competing interests:** The authors declare that they have no conflict of interest.

**Code/ Data Availability:** Not Applicable

**Acknowledgments:** The authors would like to acknowledge the financial support guaranteed through the Cooperation Agreement PFK PhD program 2019-2022 "Partnership for Knowledge-Platform 2: Health and WASH (WAter Sanitation and good Hygiene)" of AICS-Italian Agency for Development Cooperation to attend higher education programs in Italy in favor of non-Italian citizens. The authors are grateful to the Pakistan Meteorological Department (PMD) and Water and Power Development Authority (WAPDA) for sharing the data.

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
