# Peer review of "Spatio-temporal Evolution of Wet-Dry Event Features and their Transition across Upper Jhelum Basin (UJB)-South Asia"

_Natural Hazards and Earth System Sciences, 2021_

## Referee Comment (RC1)

**General Comments**

In this paper authors investigate the evolution of wet and dry events collectively in space and time over Upper Jhelum Basin for a period of 1981-2014. They use SPEI index calculated from distribution mapping based corrected ERA5 precipitation estimates and observed temperature data, and locate the hotspot regions for wet, dry and both wet-dry rapid transit events. The idea of the analysis is interesting and the potential for the results is high, however the manuscript remains mostly descriptive.

The paper is well written, with a clear, fluent and concise language and a well organised structure. I think that the manuscript can provide new insights into understanding the evolution of compound extreme events. Hence, my assessment of the manuscript is overall positive. However, some revision is needed before the work can be accepted for publication in the journal. Below detailed comments are listed:

**Minor comments:**

**C1**: Figure 1 is not well explained. I suggest that the authors should revise the figure by showing name or number of the gauging stations. I suggest to present a detail figure of study area.

**C2**: The writing and English need thorough polishing. Numerous grammatical and rhetorical issues too.

**C3**. I have some concerns about the introduction section. I think that if the authors wish this paper is well considered by experts, more attention should be devoted to discuss the extreme events in the area. Moreover, this section is lacking clarity and sufficient motivations. I suggest to improve it or better explain with realistic examples. Kindly go through the Zaman et al (2020) for extreme events in the UIB.

**Zaman, M.;** Ahmad, I.; Usman, M.; Saifullah, M.; Anjum, M.N.; Khan, M.I.; Uzair Qamar, M. Event-Based Time Distribution Patterns, Return Levels, and Their Trends of Extreme Precipitation across Indus Basin. *Water* **2020**, *12*, 3373

**C4:** As the data use to carry out a research work is the base of a research work and the most important ingredient. The authors have not provided any detail of the data they have used to carry out their work. I suggest that the authors must provide the complete detail of the data they have used in this research work. Moreover, the authors have applied any homogeneity test on the data to ensure the data quality? In data description section the authors did not mention from where they took observed data and what is the ethnicity of the data. I suggest the authors to go through the Zaman et al 2020 for the data quality and presentation.

**Zaman, M.;** Ahmad, I.; Usman, M.; Saifullah, M.; Anjum, M.N.; Khan, M.I.; Uzair Qamar, M. Event-Based Time Distribution Patterns, Return Levels, and Their Trends of Extreme Precipitation across Indus Basin. *Water* **2020**, *12*, 3373

**C5:** Line 138, I would strongly suggest adding 2-3 sentences why authors prefer to use distribution mapping method of bias correction of ERA5 precipitation and which frequency distribution was employed/fitted to the precipitation data.

**C6:** Authors used gridded ERA5 precipitation and observed temperature based potential evapotranspiration for the calculation of SPEI index. Would you please just clarify the reason why authors use gridded and observed data combination instead of use only gridded or observed datasets for both variables?

**C7:** From line 152 onward. Overall, the explanation of SPEI is very easy to understand and I think it should not be substituted by merely a reference to another publication. However, would it be possible to add basic equations to guide some type of readers?

**C8:** The authors used monthly time scale to detect floods and flash droughts. What do you mean by flash drought? Please explicitly define somewhere in manuscript.

**C9:** Figure 8, what are the units of transition time? Kindly mention it.

**C10:** Geographical coordinates are provided in figure 7 only. It would be better to add geographical coordinates to all figures or remove it from figure 7.

**C11:** Rapid transition of wet-to-dry or dry-to-wet event refers to the one extreme event is followed by the opposite event. It must not necessarily happen with similar severity level.

**C12:** Line 261-263, rephrase the sentence.

---

## Author Comment (AC1)

**Responses to Referee (1)**

The referee **comments** are highlighted in **black** and numbered with **C1-C12**, whereas the **responses** are in **red**.

**Update:**

Initially we did not provide any supplementary material along with submitted manuscript. However, considering the concerns of reviewers about the ERA5 precipitation bias correction, we added evidence of a few results, relevant to assess the reliability of ERA5 precipitation bias correction, as supplementary material. Here we present the results of four different bias correction approaches (LS-linear scaling, LOCI-local scaling intensity, PT-power transformation and DM-distribution mapping) in terms of some statistical terms. The detailed results of these bias correction approaches with respect to extreme precipitation indices are under review.

Kindly see figure S1 below as supplementary material.

[Figure]

**Figure S1:** Comparison of raw and bias corrected ERA5 precipitation statistics with observed data for the period 1981-2014

**General Comments**

In this paper authors investigate the evolution of wet and dry events collectively in space and time over Upper Jhelum Basin for a period of 1981-2014. They use SPEI index calculated from distribution mapping based corrected ERA5 precipitation estimates and observed temperature data, and locate the hotspot regions for wet, dry and both wet-dry rapid transit events. The idea of the analysis is interesting and the potential for the results is high, however the manuscript remains mostly descriptive.

The paper is well written, with a clear, fluent and concise language and a well-organized structure. I think that the manuscript can provide new insights into understanding the evolution of compound extreme events. Hence, my assessment of the manuscript is overall positive. However, some revision is needed before the work can be accepted for publication in the journal. Below detailed comments are listed:

**Response:** We would like to thank Dr. Muhammad Zaman for his fair and thorough review. Below, we give a comment-by-comment response, indicating the changes we plan to make to the manuscript.

**Specific comments:**

**C1**: Figure 1 is not well explained. I suggest that the authors should revise the figure by showing name or number of the gauging stations. I suggest presenting a detail figure of study area.

**Response:** Figure 1 was updated by incorporating names of each station.

[Figure]

**Figure 1:** Location of the UJB and spatial distribution of climatic stations

**C2**: The writing and English need thorough polishing. Numerous grammatical and rhetorical issues too.

**Response:** The text of the paper was further checked to remove grammar errors and typos.

**C3**. I have some concerns about the introduction section. I think that if the authors wish this paper is well considered by experts, more attention should be devoted to discuss the extreme events in the area. Moreover, this section is lacking clarity and sufficient motivations. I suggest to improve it or better explain with realistic examples. Kindly go through the Zaman et al (2020) for extreme events in the UIB.

**Zaman, M.;** Ahmad, I.; Usman, M.; Saifullah, M.; Anjum, M.N.; Khan, M.I.; Uzair Qamar, M. Event-Based Time Distribution Patterns, Return Levels, and Their Trends of Extreme Precipitation across Indus Basin. Water **2020**, 12, 3373

**Response:** Introduction part was revised to improve clarity and paper motivation. Moreover, the mentioned study is indeed relevant and a reference to it was added in the Introduction chapter of the revised paper.

**C4:** As the data use to carry out a research work is the base of a research work and the most important ingredient. The authors have not provided any detail of the data they have used to carry out their work. I suggest that the authors must provide the complete detail of the data they have used in this research work. Moreover, the authors have applied any homogeneity test on the data to ensure the data quality? In data description section the authors did not mention from where they took observed data and what is the ethnicity of the data. I suggest the authors to go through the Zaman et al 2020 for the data quality and presentation.

**Zaman, M.;** Ahmad, I.; Usman, M.; Saifullah, M.; Anjum, M.N.; Khan, M.I.; Uzair Qamar, M. Event-Based Time Distribution Patterns, Return Levels, and Their Trends of Extreme Precipitation across Indus Basin. Water **2020**, 12, 3373

**Response:** Suggestions were accounted for in the text incorporated under the heading "**Data Description".** Kindly see the updated version of subchapter below:

**Data Description**

The daily observed precipitation and temperature data of 15 climatic stations located within the political boundary of Pakistan were collected from Pakistan Meteorological Department (PMD) and Water and Power Development Authority (WAPDA). For the Indian side region, Indian Meteorological Department (IMD) daily gridded precipitation and temperature datasets, derived from a dense network of meteorological stations for the Indian mainland (Pai et al., 2015), were extracted at five stations and used for that region. The analysis was carried out for a period of 34 years (1981-2014), due to the availability of observed data.

In fact there are only a few climatic stations where data are available starting from 1971, but the number of stations would not be enough for the spatial analysis. The observed temperature data was used to calculate potential evapotranspiration (PET) using the Thornthwaite equation (Thornthwaite, 1948) due to data limitation. A study conducted by Beguería et al. (2014) compared the SPEI values calculated with three different methods (Penman-Manteith, Hargreaves, and Thornthwaite) and found small differences in humid regions. Mavromatis (2007) also reported similar outcomes of PET methods for drought indices calculation. Afterwards PET values were interpolated at 0.25° using Kriging with External Drift (KED), considering elevation as a predictor (Goovaerts, 2000). For the precipitation, contrasting reviews are reported in the literature about the performance of KED technique. For instance, (Masson et al., 2014) reported considerable improvement in interpolation accuracy with KED compared to other linear regressions not accounting for any predictor in high mountainous regions. On the other hand, (Berndt and Haberlandt, 2018, Ly et al., 2011) argue that topographical impact was indispensable for only temperature reconstruction at all temporal resolutions and station densities, but its influence was less clear for daily to monthly precipitation. Furthermore, all spatial interpolation techniques can perform poorly in regions with insufficient high-elevation data, due to inaccurate estimation of local lapse rates (Ruelland and Sciences, 2020). Therefore, the distribution mapping (DM)-corrected ERA5 precipitation estimates (0.25° horizontal resolution) were used in the present study. ERA5 is a relatively new reanalysis launched by European Centre for Medium-Range Weather Forecasts (ECMWF) (Saha et al., 2010). The data are developed by using advanced 4Dvar assimilation scheme and provide various atmospheric variables at 139 pressure levels for the period 1979-present time. The DM method adjusts the cumulative distribution function (CDF) of modelled precipitation to match with the observed precipitation CDF using a transfer function (Sennikovs and Bethers, 2009) and it is commonly used to correct the systematic distributional biases (Cannon et al., 2015). The Gamma distribution (Thom, 1958) with a shape and a scale parameter was found to be suitable for the precipitation distribution in the study region (Azmat et al., 2018). The suitability of ERA5 precipitation and bias correction method with respect to extreme precipitation analysis was checked against observed station data and a few results of the reliability check of DM-corrected ERA5 is provided in supplementary material.

**C5:** Line 138, I would strongly suggest adding 2-3 sentences why authors prefer to use distribution mapping method of bias correction of ERA5 precipitation and which frequency distribution was employed/fitted to the precipitation data.

**Response:** Suggestions were accounted for in the text incorporated under the heading "**Data Description".** Kindly see the updated version of subchapter "**Data Description"** in the response of C4.

**C6:** Authors used gridded ERA5 precipitation and observed temperature based potential evapotranspiration for the calculation of SPEI index. Would you please just clarify the reason why authors use gridded and observed data combination instead of use only gridded or observed datasets for both variables?

**Response:** Reviewer concern was accounted for in the text incorporated under the heading "**Data Description**". Kindly see the updated version of subchapter "**Data Description**" in the response of C4.

**C7:** From line 152 onward. Overall, the explanation of SPEI is very easy to understand and I think it should not be substituted by merely a reference to another publication. However, would it be possible to add basic equations to guide some type of readers?

**Response:** More explanation of SPEI with equations was added under the heading **"Wet and Dry Events Identification"**. Kindly find the additional explanation of SPEI below:

**Wet and Dry Events Identification**

The calculation procedure of SPEI involves two steps: fitting a log-logistic distribution to the monthly climatic water balance (P-PET) time series and then transforming the cumulative probability of the fitted distribution to a standard normal distribution (with mean equal to 0 and variance equal to 1). Accordingly, in the first step the log-logistic probability distribution function, expressed as:

$$F(x) = [1 + (\frac{\alpha}{x - \gamma})^{\beta}]^{-1}$$

where α, β and γ are the shape, scale, and origin parameters respectively, was fit to the variable x (monthly climatic water balance). In the second step, SPEI was calculated as the standardized value of F(x) as follows:

$$SPEI = W - \frac{C_o + C_1 W + C_2 W^2}{1 + d_1 W + d_2 W^2 + d_3 W^3}$$

where

$$W = \sqrt{-2\ln(F(x))} \qquad \text{for F(x)} < 0.5$$

$$W = \sqrt{-2\ln(1 - F(x))} \qquad \text{for F(x)} > 0.5$$

The parameters C0, C1, C2, d1, d2, d3 are SPEI constants (Vicente-Serrano et al., 2010). The log-logistic distribution for SPEI calculation was used and recommended by many researchers (Ullah et al., 2021a, Akhtar et al., 2020, Himayoun and Roshni, 2019, Vicente-Serrano et al., 2010). The detailed description of the SPEI calculation procedure can be found in (Vicente-Serrano et al., 2010).

**C8:** The authors used monthly time scale to detect floods and flash droughts. What do you mean by flash drought? Please explicitly define somewhere in manuscript.

**Response:** The flash drought is a relatively new type of drought. Currently, there is not a universally accepted definition or criteria for flash drought, though there is a general consensus on the principle of rapid onset or intensification characterized by moisture deficits and abnormally high temperatures for a period lasting at least 3 weeks (Lisonbee et al. 2021, Otkin et al. 2018, Hunt et al. 2009).

We incorporated this definition in the manuscript.

**C9:** Figure 8, what are the units of transition time? Kindly mention it.

**Response:** Units were added in figure 8.

**C10:** Geographical coordinates are provided in figure 7 only. It would be better to add geographical coordinates to all figures or remove it from figure 7.

**Response:** Thank you for your suggestion. Figure 7 was updated/revised.

**C11:** Rapid transition of wet-to-dry or dry-to-wet event refers to the one extreme event is followed by the opposite event. It must not necessarily happen with similar severity level.

**Response:** Yes, the rapid transition refers to the consecutive events/months of different types (one type of event followed by another type of event), regardless of their severity level. These consecutive opposite events could be of the same or of different severity level.

**C12:** Line 261-263, rephrase the sentence.

**Response:** We rephrased the sentence to make it clearer.

---

## Author Comment (AC2)

**Responses to Anonymous Referee (2)**

The referee **comments** are highlighted in **black** whereas the **responses** are in **red**.

**Update:**

Initially we did not provide any supplementary material along with submitted manuscript. However, considering the concerns of reviewers about the ERA5 precipitation bias correction, we added evidence of a few results, relevant to assess the reliability of ERA5 precipitation bias correction, as supplementary material. Here we present the results of four different bias correction approaches (LS-linear scaling, LOCI-local scaling intensity, PT-power transformation and DM-distribution mapping) in terms of some statistical terms. The detailed results of these bias correction approaches with respect to extreme precipitation indices are under review.

Kindly see figure S1 below as supplementary material.

[Figure]

**Figure S1:** Comparison of raw and bias corrected ERA5 precipitation statistics with observed data for the period 1981-2014

**General Comments**

The paper of Ansari and Grossi provides an exercise where the main features of dry-wet condition transitions are analysed at the monthly time scale in the Upper Jhelum Basin located in between India and Pakistan. The authors use a mixed dataset for the period 1981-2014, including ERA5 derived precipitation and observed temperature, they first calculate the SPEI index and then derive several related indices highlighting both dry, wet and combined dry/wet transition events characteristics.

The main contribution of the paper, besides the specific results achieved for the study area, is the effort of proposing a methodological framework, yet based on well-known approaches and methods. I suggest some improvements detailed below. I hope my comments can contribute to enhancing the quality of the paper.

**Response:** We would like to thank Anonymous Referee #2 for the fair and thorough review. Below, we give a comment-by-comment response, indicating the changes we plan to make to the manuscript.

**Specific comments**

- First, I suggest the authors carefully checking the text to avoid several grammar errors and typos widespread in the manuscript (I list some of them at the end of the review as examples).

**Response:** The text of the paper was further checked to remove grammar errors and typos.

- I classify this comment as 'main' because it concerns the title. In practice, if the authors agree, it can be easily solved. I don't agree with the term "wet event" because the expectation of the general audience is for smaller time scales than monthly. Therefore, for the sake of clarity, I suggest different phrasing. Probably "wet-dry months" is a correct, yet simple choice (please refer also to the note at lines 226-227).

**Response:** We agree with your point of view. However, the manuscript primarily works on wet and dry events rather than floods and droughts. In the text we mention the clear difference between flood and wet event (kindly check the LL 77-80 of the preprint) and support the results with the historical flood and drought events occurred in the region. We also clearly explain the meaning we give to these terms (flood-drought and wet-dry event) (please refer also to the note at LL 77-78 of the preprint). If the Reviewer still thinks we should change the term in the title, we will do it.

- I see several problems with data. First, I can't read the source of observed temperatures. Then, the reliability of ERA5 precipitation data needs to be accurately checked against available observations. In this regard, the authors provide a reference to a conference abstract (Ansari and Grossi, 2021). It's not enough, a section about data validation is needed. Finally, I'm not that keen

on using the Thornthwaite method, which is very dated. I would suggest using at least a temperature-based model, e.g. Hargreaves-Samani. However, ERA5 provides potential evaporation data, a comparison between such data and the results achieved by the authors with another method would be interesting and could provide useful insights. The authors should discuss their choice of relying partially on datasets and partially on ground observations.

**Response:** Reviewer's concerns have been discussed and incorporated under the heading **"Data Description".** Moreover, a few results of the reliability check of DM-corrected ERA5 is now provided as supplementary material. The detailed evaluation of different gridded precipitation datasets and different bias correction methods with respect to extreme precipitation indices is under review.

Kindly see the revised version of **Data Description** below. Hopefully it clarifies any aspect of data origin and their usage.

**Data Description**

The daily observed precipitation and temperature data of 15 climatic stations located within the political boundary of Pakistan were collected from Pakistan Meteorological Department (PMD) and Water and Power Development Authority (WAPDA). For the Indian side region, Indian Meteorological Department (IMD) daily gridded precipitation and temperature datasets, derived from a dense network of meteorological stations for the Indian mainland (Pai et al., 2015), were extracted at five stations and used for that region. The analysis was carried out for a period of 34 years (1981-2014), due to the availability of observed data. In fact there are only a few climatic stations where data are available starting from 1971, but the number of stations would not be enough for the spatial analysis. The observed temperature data was used to calculate potential evapotranspiration (PET) using the Thornthwaite equation (Thornthwaite, 1948) due to data limitation. A study conducted by Beguería et al. (2014) compared the SPEI values calculated with three different methods (Penman-Manteith, Hargreaves, and Thornthwaite) and found small differences in humid regions. Mavromatis (2007) also reported similar outcomes of PET methods for drought indices calculation. Afterwards PET values were interpolated at $0.25^o$ using Kriging with External Drift (KED), considering elevation as a predictor (Goovaerts, 2000). For the precipitation, contrasting reviews are reported in the literature about the performance of KED technique. For instance, (Masson et al., 2014) reported considerable improvement in interpolation accuracy with KED compared to other linear regressions not accounting for any predictor in high mountainous regions. On the other hand, (Berndt and Haberlandt, 2018, Ly et al., 2011) argue that topographical impact was indispensable for only temperature reconstruction at all temporal resolutions and station densities, but its influence was less clear for daily to monthly precipitation. Furthermore, all spatial interpolation techniques can perform poorly in regions with insufficient high-elevation data, due to inaccurate estimation of local lapse rates (Ruelland and Sciences, 2020). Therefore, the distribution mapping (DM)-corrected ERA5 precipitation estimates ($0.25^o$

horizontal resolution) were used in the present study. ERA5 is a relatively new reanalysis launched by European Centre for Medium-Range Weather Forecasts (ECMWF) (Saha et al., 2010). The data are developed by using advanced 4Dvar assimilation scheme and provide various atmospheric variables at 139 pressure levels for the period 1979-present time. The DM method adjusts the cumulative distribution function (CDF) of modelled precipitation to match with the observed precipitation CDF using a transfer function (Sennikovs and Bethers, 2009) and it is commonly used to correct the systematic distributional biases (Cannon et al., 2015). The Gamma distribution (Thom, 1958) with a shape and a scale parameter was found to be suitable for the precipitation distribution in the study region (Azmat et al., 2018). The suitability of ERA5 precipitation and bias correction method with respect to extreme precipitation analysis was checked against observed station data and a few results of the reliability check of DM-corrected ERA5 is provided in supplementary material.

- Overall, I found the results and, mainly, the discussion, not particularly vivid. The authors should strive to emphasize better the added value of their study, avoiding not very fitting comments. E.g., I don't think the sentence in LL396-398 is very appropriate, because it refers to actual ET, while the method used refers to potential ET (PET).

 **Response:** Efforts have been made to improve this section.

Regarding the LL396-398, authors intended to highlight the link between global warming and drought conditions, along with the provided citation. Even if the mentioned sentence refers to actual ET, PET is indeed the upper limit of actual ET.  We rephrased the mentioned sentence to make it clear.

**Minor comments**

L30: the authors refer to AR5, maybe they can update considering the brand new AR6

**Response:** the manuscript was changed according to the suggestion. Reference to the climate change projections for South Asia in AR6 was added.

LL80-85: I think this sentence should be better placed in the Conclusions

**Response:** The manuscript was modified accordingly.

L93: SSI is cited only here and not explained

**Response:** SSI stands for standardized streamflow index. This piece of information was added in the manuscript.

L119: a paper under review is cited. I would avoid it. Anyway, it is not in the References

**Response:** Authors removed this citation.

Fig. 1: it is not very clear. Only part of the Kunhar borders is visible. Please flip the colour palette of Elevation (high brown and low green)

**Response:** Figure 1 was updated to make it clearer

[Figure]

**Figure 1:** Location of the UJB and spatial distribution of climatic stations

L136: basically, a period of 35 years is not enough for such kind of analysis. Please extend the discussion of this issue and hint at the possibility of using an extended (in the past) ERA5 dataset

**Response:** Yes, authors acknowledge the reviewer's point of view. Availability of observed data is the main limitation in this regard. There are only a few climatic stations where data are available from 1971, but the number of stations would not be enough for the correction of ERA5 precipitation and interpolation of observed temperature.

Discussion about the time period selection for the analysis was added under the heading "Data Description.

Table 1 and elsewhere: I guess it's "extremely wet", "severely wet", etc., not "extreme wet", "severe wet", etc.

**Response:** We actually meant to use two cumulative (paired) adjectives (extreme wet/severe wet) rather than an adverb (Extremely/severely)+ an adjective, as both forms are used in English. We prefer the shorter and more effective form.

Section 4.4: I suppose that also the number of transitions for each grid cell should be considered. Is it so? If not, why?

**Response:** A figure showing the number of transitions for each grid cell was incorporated into the manuscript. Kindly see figure 8.

[Figure]

**Figure 8:** Number of transitions for wet-to-dry (left) and dry-to-wet (right) events for the three levels of severity (moderate, severe, extreme) for the period 1981-2014

L200: alteration --> maybe "rapid transition"?

**Response:** Change was made.

Fig.3: The year 1980 should not appear here, it's not within the analyzed period

**Response:** Figure 3 was updated/revised.

[Figure]

**Figure 3:** Annual variations in the number of months affected by wet/dry conditions during the 1981-2014 period. The brown and blue colors present dry and wet months, respectively. Different shades of the colors define the different severity levels (EW-wet, ED-extreme dry, SW-severe wet, SD-severe dry, MW-moderate wet, MD-moderate dry)

Fig. 4: it's like AWD and ADD, and MWD and MDD are almost complementary (my feeling)

**Response:** Thank you for your valuable comments. Authors acknowledge your feelings and added few more lines considering your suggestions.

L279: TDI results are not yet introduced

**Response:** The text was changed to account for your observation

Fig. 7: only here maps coordinates are made explicit. Please make all maps homogeneous.

**Response:** Thank you for your suggestion. Figure 7 was updated to make it homogeneous with others of the same type. Coordinates are shown in Figure 1.

L328 and L339: "a greater number": please quantify

**Response:** Quantification was added

Fig. 8: what are the units? Months?

**Response:** Units were added in figure 8.

**Typos and English grammar (examples)**

L8: "more than" or "rather than"

L24: Extremes weather events

L29: extremes events

L67: standardized indices, which facilitates

L123: The monsoon pattern bring

L161: The severity levels… was classified

L167: Following to Spinoni et al.

L215: not clear, please rephrase

L226: the terms… presents…

L266: "…exhibit two distinct parts of the basin". Not clear, please rephrase

L313: The higher positive values: I guess "the highest". Also, in the next line, "highest"

L384: El Nino suppress monsoon rainfall activity over Pakistan

**Response:** The text of the paper was further checked to remove grammar errors and typos.

---

## Author Response (AR1)

| First Referee Comment (RC1) | Author Comment (AC) |
|---|---|
| In this paper authors investigate the evolution of wet and dry events collectively in space and time over Upper Jhelum Basin for a period of 1981-2014. They use SPEI index calculated from distribution mapping based corrected ERA5 precipitation estimates and observed temperature data, and locate the hotspot regions for wet, dry and both wet-dry rapid transit events. The idea of the analysis is interesting and the potential for the results is high, however the manuscript remains mostly descriptive.
 The paper is well written, with a clear, fluent and concise language and a well-organized structure. I think that the manuscript can provide new insights into understanding the evolution of compound extreme events. Hence, my assessment of the manuscript is overall positive.
 However, some revision is needed before the work can be accepted for publication in the journal. Below detailed comments are listed: | We would like to thank Dr. Muhammad Zaman for his fair and thorough review. Below, we give a comment-by-comment response, indicating the changes we made in the revised manuscript. |
| **C1**: Figure 1 is not well explained. I suggest that the authors should revise the figure by showing name or number of the gauging stations. I suggest presenting a detail figure of study area. | Figure 1 was updated accordingly. |
| **C2**: The writing and English need thorough polishing. Numerous grammatical and rhetorical issues too. | The text of the paper was further checked to remove grammar errors and typos. |
| **C3**. I have some concerns about the introduction section. I think that if the authors wish this paper is well considered by experts, more attention should be devoted to discuss the extreme events in the area. Moreover, this section is lacking clarity and sufficient motivations. I suggest to improve it or better explain with realistic examples. Kindly go through the Zaman et al (2020) for extreme events in the UIB.
 **Zaman, M.;** Ahmad, I.; Usman, M.; Saifullah, M.; Anjum, M.N.; Khan, M.I.; Uzair Qamar, M. Event-Based Time Distribution Patterns, Return Levels, and Their Trends of Extreme Precipitation across Indus Basin. Water **2020**, 12, 3373 | Introduction part was revised to improve clarity and paper motivation. Moreover, the mentioned study is indeed relevant and a reference to it was added in the Introduction chapter of the revised paper. |
| **C4:** As the data use to carry out a research work is the base of a research work and the most important ingredient. The authors have not provided any detail of the data they have used to carry out their work. I suggest that the authors must provide the complete detail of the data they have used in this research work. Moreover, the authors have applied any homogeneity test on the data to ensure the data quality? In data description section the authors did not mention from where they took observed data and what is the ethnicity of the data. I suggest the authors to go through the Zaman et al 2020 for the data quality and presentation.
 **Zaman, M.;** Ahmad, I.; Usman, M.; Saifullah, M.; Anjum, M.N.; Khan, M.I.; Uzair Qamar, M. Event-Based Time Distribution Patterns, Return Levels, and Their Trends of Extreme Precipitation across Indus Basin. Water **2020**, 12, 3373 | Suggestions were accounted for in the text incorporated under the heading "**Data Description".** Kindly see the updated version of the manuscript. |
| **C5:** Line 138, I would strongly suggest adding 2-3 sentences why authors prefer to use distribution mapping method of bias correction of ERA5 precipitation and which frequency distribution was employed/fitted to the precipitation data. | Suggestions were accounted for in the text incorporated under the heading "**Data Description".** Kindly see the updated version of the manuscript. |
| **C6:** Authors used gridded ERA5 precipitation and observed temperature based potential evapotranspiration for the calculation of SPEI index. Would you please just clarify the reason why authors use gridded and observed data combination instead of use only gridded or observed datasets for both variables? | Reviewer concern was accounted for in the text incorporated under the heading "**Data Description".**
 Kindly see the updated version of the manuscript. |

| | |
|---|---|
| **C7:** From line 152 onward. Overall, the explanation of SPEI is very easy to understand and I think it should not be substituted by merely a reference to another publication. However, would it be possible to add basic equations to guide some type of readers? | More explanation of SPEI with equations was added under the heading **"Wet and Dry Events Identification".**
Kindly see the updated version of the manuscript. |
| **C8:** The authors used monthly time scale to detect floods and flash droughts. What do you mean by flash drought? Please explicitly define somewhere in manuscript. | **Response:** The flash drought is a relatively new type of drought. Currently, there is not a universally accepted definition or criteria for flash drought, though there is a general consensus on the principle of rapid onset or intensification characterized by moisture deficits and abnormally high temperatures for a period lasting at least 3 weeks (Lisonbee et al. 2021, Otkin et al. 2018, Hunt et al. 2009). We incorporated this definition in the revised manuscript. |
| **C9:** Figure 8, what are the units of transition time? Kindly mention it. | Units were added in figure 8. |
| **C10:** Geographical coordinates are provided in figure 7 only. It would be better to add geographical coordinates to all figures or remove it from figure 7. | Thank you for your suggestion. Figure 7 was updated. |
| **C11:** Rapid transition of wet-to-dry or dry-to-wet event refers to the one extreme event is followed by the opposite event. It must not necessarily happen with similar severity level. | Yes, the rapid transition refers to the consecutive events/months of different types (One type of event followed by another type of event), regardless of their severity level. These consecutive opposite events could be of the same or of different severity level. |
| **C12:** Line 261-263, rephrase the sentence. | We rephrased the sentence to make it clearer. |

| Second Referee Comment (RC2) | Author Comment (AC) |
|---|---|
| The paper of Ansari and Grossi provides an exercise where the main features of dry-wet condition transitions are analysed at the monthly time scale in the Upper Jhelum Basin located in between India and Pakistan. The authors use a mixed dataset for the period 1981-2014, including ERA5 derived precipitation and observed temperature, they first calculate the SPEI index and then derive several related indices highlighting both dry, wet and combined dry/wet transition events characteristics.
The main contribution of the paper, besides the specific results achieved for the study area, is the effort of proposing a methodological framework, yet based on well-known approaches and methods. I suggest some improvements detailed below. I hope my comments can contribute to enhancing the quality of the paper. | We would like to thank Anonymous Referee #2 for the fair and thorough review.
Below, we give a comment-by-comment response, indicating the changes we plan to make to the manuscript. |
| First, I suggest the authors carefully checking the text to avoid several grammar errors and typos widespread in the manuscript (I list some of them at the end of the review as examples). | The text of the paper was further checked to remove grammar errors and typos. |
| I classify this comment as 'main' because it concerns the title. In practice, if the authors agree, it can be easily solved. I don't agree with the term "wet event" because the expectation of the general audience is for smaller time scales than monthly. Therefore, for the sake of clarity, I suggest different phrasing. Probably "wet-dry months" is a correct, yet simple choice (please refer also to the note at lines 226-227). | We agree with your point of view. However, the manuscript primarily works on wet and dry events rather than floods and droughts. In the text we mention the clear difference between flood and wet event (kindly check the LL 77-80 of the preprint) and support the results with the historical flood and drought events occurred in the region. We also clearly explain the meaning we give to these terms (flood-drought and wet-dry event) (please refer also to the note at LL 77-78 of the preprint). If the Reviewer still thinks we should change the term in the title, we will do it. |
| I see several problems with data. First, I can't read the source of observed temperatures. Then, the reliability of ERA5 precipitation data needs to be accurately checked against available observations. In this regard, the authors provide a reference to a conference abstract (Ansari and Grossi, 2021). It's not enough, a section about data validation is needed. Finally, I'm not that keen on using the Thornthwaite method, which is very dated. I would suggest using at least a temperature-based model, e.g. Hargreaves-Samani. However, ERA5 provides potential evaporation data, a comparison between such data and the results achieved by the authors with another method would be interesting and could provide useful insights. The authors should discuss their choice of relying partially on datasets and partially on ground observations | Reviewer's concerns have been discussed and incorporated under the heading **"Data Description".** Moreover, a few results of the reliability check of DM-corrected ERA5 is now provided as supplementary material. The detailed evaluation of different gridded precipitation datasets and different bias correction methods with respect to extreme precipitation indices is under review.
Kindly see the **Data Description** heading in the revised version of the manuscript. Hopefully it clarifies any aspect of data origin and their usage. |
| Overall, I found the results and, mainly, the discussion, not particularly vivid. The authors should strive to emphasize better the added value of their study, avoiding not very fitting comments. E.g., I don't think the sentence in LL396-398 is very appropriate, because it refers to actual ET, while the method used refers to potential ET (PET). | Efforts have been made to improve this section.
Regarding the LL396-398, authors intended to highlight the link between global warming and drought conditions, along with the provided citation. Even if the mentioned sentence refers to |

| | actual ET, PET is indeed the upper limit of actual ET. We rephrased the mentioned sentence to make it clear. |
|---|---|
| **Minor comments** | |
| L30: the authors refer to AR5, maybe they can update considering the brand new AR6 | Reference to the climate change projections for South Asia in AR6 was added. |
| LL80-85: I think this sentence should be better placed in the Conclusions | The manuscript was modified accordingly. |
| L93: SSI is cited only here and not explained | SSI stands for standardized streamflow index. This piece of information was added in the revised manuscript. |
| L119: a paper under review is cited. I would avoid it. Anyway, it is not in the References | We removed this citation. |
| Fig. 1: it is not very clear. Only part of the Kunhar borders is visible. Please flip the colour palette of Elevation (high brown and low green) | Figure 1 was updated accordingly to make it clearer. See figure 1 in revised manuscript. |
| L136: basically, a period of 35 years is not enough for such kind of analysis. Please extend the discussion of this issue and hint at the possibility of using an extended (in the past) ERA5 dataset | Yes, authors acknowledge the reviewer's point of view. Availability of observed data is the main limitation in this regard. There are only a few climatic stations where data are available from 1971, but the number of stations would not be enough for the correction of ERA5 precipitation and interpolation of observed temperature. Discussion about the time period selection for the analysis was added under the heading "Data Description". |
| Table 1 and elsewhere: I guess it's "extremely wet", "severely wet", etc., not "extreme wet", "severe wet", etc. | We meant to use two cumulative (paired) adjectives (extreme wet/severe wet) rather than an adverb (Extremely/severely) + an adjective, as both forms are used in English. We prefer the shorter and more effective form. |
| Section 4.4: I suppose that also the number of transitions for each grid cell should be considered. Is it so? If not, why? | A figure showing the number of transitions for each grid cell was incorporated into the revised manuscript. Kindly see figure 8. |
| L200: alteration --> maybe "rapid transition"? | Change was made. |
| Fig.3: The year 1980 should not appear here, it's not within the analyzed period | Figure 3 was updated. |
| Fig. 4: it's like AWD and ADD, and MWD and MDD are almost complementary (my feeling) | Thank you for your valuable comments. Authors acknowledge your feelings and add few more lines considering your suggestions. |
| L279: TDI results are not yet introduced | The text was changed to account for your observation |
| Fig. 7: only here maps coordinates are made explicit. Please make all maps homogeneous. | Thank you for your suggestion. Figure 7 was updated to make it homogeneous with others of the same type. Coordinates are shown in Figure 1 only. |
| L328 and L339: "a greater number": please quantify | Quantification was added |

| | |
|---|---|
| Fig. 8: what are the units? Months? | Units were added in figure 8. |
| **Typos and English grammar (examples)** | The text of the paper was further checked to remove grammar errors and typos. |